# Federated Learning for Appearance-based Gaze Estimation in the Wild

**Mayar Elfares[1,2] Zhiming Hu[1,3] Pascal Reisert[2] Andreas Bulling[1] Ralf Küsters[2]**

[1] *Institute for Visualisation and Interactive Systems,* [2] *Institute of Information Security,*

[3] *Institute for Modelling and Simulation of Biomechanical Systems*

*University of Stuttgart, Germany*

{MAYAR.ELFARES, ZHIMING.HU, ANDREAS.BULLING}@VIS.UNI-STUTTGART.DE

{PASCAL.REISERT, RALF.KUESTERS}@SEC.UNI-STUTTGART.DE

## Abstract

Gaze estimation methods have significantly matured in recent years, but the large number of eye images required to train deep learning models poses significant privacy risks. In addition, the heterogeneous data distribution across different users can significantly hinder the training process. In this work, we propose the first federated learning approach for gaze estimation to preserve the privacy of gaze data. We further employ pseudo-gradient optimisation to adapt our federated learning approach to the divergent model updates to address the heterogeneous nature of in-the-wild gaze data in collaborative setups. We evaluate our approach on a real-world dataset (MPIIGaze) and show that our work enhances the privacy guarantees of conventional appearance-based gaze estimation methods, handles the convergence issues of gaze estimators, and significantly outperforms vanilla federated learning by 15.8% (from a mean error of 10.63 degrees to 8.95 degrees). As such, our work paves the way to develop privacy-aware collaborative learning setups for gaze estimation while maintaining the model's performance.

**Keywords:** Gaze estimation, federated learning, privacy, gaze data distribution

## 1. Introduction

Human eye gaze is a crucial non-verbal cue used in a wide variety of applications, such as gaze-contingent rendering in virtual reality (Hu et al., 2019, 2020b, 2021; Hu, 2020; Hu et al., 2020a), gaze-based interaction (Mardanbegi et al., 2019; Piumsomboon et al., 2017), gaze-assisted collaboration (Higuch et al., 2016; Zhang et al., 2017c), as well as eye movement-based task recognition (Hu et al., 2022; Coutrot et al., 2018). Given the importance of eye gaze, many researchers have focused on the problem of gaze estimation (Baluja and Pomerleau, 1993; Liang et al., 2013; Choi et al., 2013; Lu et al., 2014), i.e. estimating gaze position or direction from eye images. However, the collection of the large amounts of eye images required to train deep learning models, or the exchange of such data across networks, can pose significant privacy risks. In addition, the heterogeneous data distribution across different users in real-world settings (in-the-wild settings) can significantly hinder the training process of gaze estimation methods (Zhang et al., 2015, 2018). Therefore, preserving privacy and maintaining high performance for heterogeneous data become two main challenges for gaze estimation.

To address these challenges, recent works have developed privacy-preserving approaches for gaze applications. Nonetheless, they either only handle specific attacks and data vulnerabilities (Bozkir et al., 2021; Steil et al., 2019; Hagestedt et al., 2020), or mainly focus on enhancing data privacy, at the expense of model performance (Li et al., 2021; Bozkir et al., 2021; Liu et al., 2019; Steil et al., 2019).

In this work, we propose the first federated learning (FL) approach (McMahan et al., 2017) for gaze estimation. Federated learning is a machine learning paradigm that enables training algorithms across multiple local datasets without exchanging data samples, in order to alleviate the data sharing privacy risks. In addition, we employ pseudo-gradient optimisation to adapt our federated learning approach to the divergent model updates caused by the heterogeneous gaze data distribution among users. We evaluate our approach on the MPIIGaze dataset (Zhang et al., 2017b) where the privacy-sensitive nature, as well as the heterogeneous distribution of in-the-wild gaze data, is prominent. Our experimental results show that our work alleviates the privacy concerns due to data sharing of conventional gaze estimation approaches. It also reduces the model updates divergence in collaborative setups while significantly outperforming vanilla federated learning by 15.8% (from a mean error of 10.63 degrees to 8.95 degrees). Finally, we discuss our method's impact on users' privacy and data heterogeneity in terms of fairness and robustness.

In summary, the main contributions of our work are as follows:

- We propose the first federated learning approach for gaze estimation to preserve the privacy of gaze data.

- We employ pseudo-gradient optimisation to adapt our federated learning approach to the divergent model updates in order to handle the heterogeneous data distribution in collaborative setups.

- We show that our approach enhances the privacy guarantees of conventional training methods, handles the convergence of gaze estimation models, and significantly outperforms vanilla federated learning.

## 2. Related Work

**Gaze Estimation Methods**   Gaze estimation methods can be generally categorised as either model-based or appearance-based (Zhang et al., 2017b). Model-based methods employ features detected from eye images to estimate gaze direction and can hardly obtain good performance in real-world settings because accurate eye feature detection relies on high-resolution images and homogeneous illumination (Zhang et al., 2017b, 2019). In contrast, appearance-based approaches directly regress gaze direction from eye images (Baluja and Pomerleau, 1993; Liang et al., 2013; Choi et al., 2013; Lu et al., 2014) and can handle low-resolution images and different gaze ranges. However, appearance-based methods require a larger number of training eye images than model-based approaches to cover the significant variability in eye appearance (Zhang et al., 2017b), which poses serious privacy risks since eye images contain ample personal information, such as gender (Sammaknejad et al., 2017), identity (Cantoni et al., 2015), and personality traits (Hoppe et al., 2018). Moreover, the heterogeneous data distribution across different users in real-world settings, which is caused

by many factors including the differences in gaze range, head pose, illumination condition, and personal appearance (Zhang et al., 2017b, 2018), can significantly hinder the training process of appearance-based gaze estimation methods.

Recent works (Li et al., 2021; Bozkir et al., 2021; Hagestedt et al., 2020; Liu et al., 2019; Steil et al., 2019; Xu et al., 2021) have developed privacy-preserving approaches for gaze applications. These works mainly focus on differential privacy solutions, where the model's performance is decreased through noise addition (Li et al., 2021; Bozkir et al., 2021; Liu et al., 2019; Steil et al., 2019), including some dataset-dependent approaches (e.g. customising added noise to the properties of the dataset) (Bozkir et al., 2021; Steil et al., 2019), while others solely reduce adversarial attacks (Hagestedt et al., 2020). In contrast, our work focuses on developing a privacy-preserving gaze estimator that maintains high estimation performance.

**Federated Learning** Federated learning (McMahan et al., 2017) is a machine learning setting where multiple decentralised edge devices (a.k.a. clients) or servers collaborate in solving a machine learning problem. Each client owns a local training dataset which is never exchanged nor transferred to the server; instead, focused updates intended for immediate aggregation are used to achieve the learning objective. These updates are ephemeral and cannot contain more information than the raw data according to the data processing inequality principle. In addition, focused collection and data minimisation principles (cit) are applied.

FL architectures (Kairouz et al., 2021) can be categorised into centralised (a.k.a. client-server) or decentralised (a.k.a. peer-to-peer) training. FL can also be classified according to the type of clients. In cross-silo FL, participants are multiple organisations (e.g., medical, financial, or geo-distributed data centres) that train on siloed data, while in cross-device FL, participants consist of a large number of mobile or IoT devices. FL can be further classified according to how data is partitioned among the clients in the feature and sample spaces. Horizontal FL applies to the scenario where the clients share overlapping data features but differ in data samples, in contrast to vertical FL. In this work, we train our gaze estimator under a cross-device centralised horizontal federated learning setting.

The federated optimisation problem (Wang et al., 2021) is different from typical distributed optimisation problems in terms of data heterogeneity, namely in the non-independent and identical distribution (non-IID) of data among clients, the imbalance in size of the local training sets, the limited and expensive communication, the computing capabilities, and the clients' availability. To effectively address these heterogeneity challenges, multi-model FL approaches were introduced (Kulkarni et al., 2020; Kairouz et al., 2021). Multi-model learning approaches in FL can be summarised as multi-task learning (Zhang and Yang, 2017) (i.e. considering each client's local problem as an independent task), local fine-tuning (Kairouz et al., 2021) (performing local training steps locally on the final model), and meta-learning (Baxter, 2000; Fallah et al., 2020; T Dinh et al., 2020) (exploiting the metadata) solutions. Furthermore, solving local updates as pseudo-gradients was first proposed for speech tasks (Chen and Huo, 2016), while adaptive learning was introduced in (Reddi et al., 2020) with the goal of incorporating knowledge from previous iterations for a better-informed optimisation. In consequence, Reddi et al. (Reddi et al., 2020) proposed FedOpt and showed that the nature of federated learning allows optimising the model on two levels, the user and server sides. In our work, we mainly focus on FedOpt, as, unlike multi-task learning, it does not

require stateful updates, which makes it suitable for cross-device setups. It also empirically outperforms meta-learning approaches (Mills et al., 2021), and can be fine-tuned. Overall, FL holds the promise of increasing usability by training gaze estimation models on large and diverse datasets of different participants while preserving data privacy.

## 3. Methodology

In this work, we propose a federated learning approach for gaze estimation that allows model training to adapt to the divergence of participants' heterogeneous data while enhancing users' privacy.

### 3.1. Gaze Data Distribution

Previous works show that gaze estimation error is significantly affected by many factors including glasses because of distortions, reflections, and thick frames, illumination conditions between indoor and outdoor environments, personal appearance changes between long-term recordings, different times of day, varying amounts of data samples per user, as well as gender, race and age differences (Zhang et al., 2017b). Earlier works (Hansen and Ji, 2009; Zhang et al., 2015; Mora and Odobez, 2013; Zhang et al., 2017a,b) tried to overcome these challenges by improving user-specific and user-independent metrics. However, they rarely consider the associated privacy risks of data-sharing and only offer one global dataset-specific model for all participants.

Formally, for a better understanding of data heterogeneity, gaze estimation represents a supervised task with features $x$ and labels $y$. The dataset contains $N$ participants with their respective datasets $\{D_i\}_{i=1}^N$. Each participant $i$ is sampled from the distribution $Q$ over all available participants. Then, a data sample $(x, y)$ is sampled from the participant's local data distribution $P_i(x, y)$.

The differences between the data distribution $P_i(x, y)$ and $P_j(x, y)$ for different participants $i$ and $j$ is referred to as non-IID. The joined probability $P_i(x, y)$ can be formulated as $P_i(y \mid x)P_i(x)$ or $P_i(x \mid y)P_i(y)$. Hence, decoupling this joint distribution allows defining the distribution skews as:

1. **Non-identical distribution**

   - **Feature distribution skew (covariate shift):** The marginal distributions $P_i(x)$ differ across participants, even if $P_i(y \mid x) = P_j(y \mid x)$ for all participants $i$ and $j$. In other words, each participant stores features that have different statistical distributions compared to other participants (e.g. different lighting conditions and personal appearance).

   - **Label distribution skew (prior probability shift):** The marginal distributions $P_i(y)$ differ across participants, even if $P_i(y \mid x) = P_j(y \mid x)$ for all participants $i$ and $j$. Each participant owns labels that have different statistical distributions compared to other participants (e.g. some gaze targets are more frequent than others).

   - **Same label, different features (concept shift):** The conditional distributions $P_i(x \mid y)$ differ across participants, even if $P_i(y) = P_j(y)$ for all participants $i$ and $j$. This means that different participants may have the same labels corresponding to

different features for each participant (e.g. the same gaze target location can be mapped to different data samples).

- **Same features, different label (concept shift):** The conditional distributions $P_i(y \mid x)$ may differ across participants, even if $P_i(y) = P_j(y)$ for all participants $i$ and $j$. This means that different participants may have the same features corresponding to different labels for each participant (e.g. the data is collected from different laptops with different cameras and screen sizes).

2. **Violation of independence** Violations of independence can appear whenever the distribution $Q$ varies during the data collection or its simulation during the training process. In practice, devices should typically satisfy eligibility requirements (e.g., devices should be idle, connected to an un-metered wi-fi connection, charged, and at night local time) in order to participate in the training process. Hence, patterns in device availability and local timing introduce strong bias in the data source. In the used datasets, the different screen times and the number of days used for data collection significantly differ across participants.

3. **Quantity imbalance** Different participants hold significantly different dataset sizes due to the heavier use of devices.

Therefore, since the heterogeneity of data distribution is prominent in gaze estimation tasks, we take into consideration the above challenges to improve the gaze estimator performance across participants in collaborative setups.

### 3.2. Gaze Estimation Model

Based on Zhang et al. (Zhang et al., 2015), a multi-modal convolutional neural network (CNN) is implemented for gaze estimation. The CNN regresses the pre-processed dataset input $x = (e, h)$, where $e$ and $h$ denote the 60x36 eye image and 1x2 head angle, respectively, to the gaze angles $\hat{g}$ in the normalised space. The final output $g$ consists of the yaw $\hat{g}_\phi$ and pitch $\hat{g}_\theta$ angles. The loss function between the predicted $\hat{g}$ and ground-truth $g$ angle vectors is calculated by the sum of the individual $L1$ losses.

In this paper, we mainly focus on the CNN optimiser that minimises the loss function and updates the model parameters $\theta$. For CNN training, stochastic gradient descent (SGD) is used with a learning rate $\eta$ of $10^{-5}$. A momentum term $\gamma$ of 0.9 is added to stabilise the optimisation, speed up convergence, and reduce oscillations. Finally, a Nesterov accelerated gradient (NAG) is added for a better anticipatory gradient update, as shown in Equation 1.

$$\theta = \theta - v_t \quad : \quad v_t = \gamma v_{t-1} + \eta \nabla_\theta J(\theta - \gamma v_{t-1}) \tag{1}$$

### 3.3. Federated Learning

Federated learning (McMahan et al., 2017) is an approach to train deep learning models on large and diverse datasets while preserving participants' privacy. We specifically study a cross-device centralised horizontal FL setup. This setup is characterised by the fact that the data distribution is massively parallel and can include a large number of devices. It also builds on a client-server architecture since a central server is needed to orchestrate the

exchange and aggregation of model updates. Finally, the setup is horizontal as participants share overlapping data features as they all train on similar modalities but differ in data samples with every participant having their own data. Formally, $X_{p_i} = X_{p_j}, Y_{p_i} = Y_{p_j}, I_{p_i} \neq I_{p_j}, \forall D_{p_i}, D_{p_j}, p_i \neq p_j$ where $X$ denotes the data feature space, $Y$ denotes the label space, and $D$ denotes the dataset. Both spaces pairs $(X_{p_i}, Y_{p_i})$ and $(X_{p_j}, Y_{p_j})$ are assumed to be the same, whereas the user identifiers $I_{p_i}$ and $I_{p_j}$ are assumed to be different.

During model training, as shown in Figure 1, each participant $p_i$ in a set of participants trains a model without exchanging their local data. The server initialises the model's hyper-parameters, weights, and biases along with the number of communication rounds and of local epochs. Thus, at each communication round $t$, the server selects a random cohort of participants $C$ for training. In this work, $C = 0.8$ to reduce the communication traffic and to mitigate the straggler effect introduced by the computational heterogeneity of the participants. The central server sends the current model parameters to the selected participants. Each selected participant samples a batch of samples $D_i^t$ from its local dataset $D_i$. The batch size is denoted by $|D_i^t|$. To train the local gaze estimation model, we minimize the non-convex objective function $f_i$:

$$\min_{w \in R^d} f(w) = \frac{1}{N} \sum_{i=1}^{N} f_i(w) \tag{2}$$

In the supervised learning setting, $f_i$ is the expected loss over the data distribution of participants $N$: $f_i(w) = E_{(x,y) \sim P_i}[l_i(w; x, y)]$ where $l_i(w; x, y)$ denotes the error of model $w$ in predicting the true gaze target $y \in Y_i$ given the input $x \in X_i$, and $P_i$ is the distribution over $X_i \times Y_i$ (the set of indexes of data samples in $D_i$) with $n_i = |P_i|$.

Since the data is non-IID, $P_i$ represents the empirical distribution of the participant's data samples. Therefore, $f_i(w) = \sum_{(x,y) \in D_i} P_i(x, y) l_i(w; x, y)$ where $P_i(x, y)$ is the probability that participant $i$ selects a particular sample $(x, y)$. Therefore, we can rewrite Equation 2 as: $\min_{w \in R^d} f(w) = \frac{1}{N} \sum_{i=1}^{N} \frac{n_i}{N} F_i(w)$ where, $F_i(w) = \frac{1}{n_i} \sum_{i=1}^{n_i} f_i(w)$.

Each participant trains a model using their respective data and sends the model weight updates to the central server. The server then combines the model updates received from all participants, typically by computing the average (e.g. FedAvg (McMahan et al., 2017)) and sends the aggregated model updates back to the participants. This cycle is repeated until the maximum number of rounds is reached. Furthermore, conventional FL optimisation techniques (e.g. SGD) often require high communication costs. As a solution, multiple local client updates are used (McMahan et al., 2017).

Nonetheless, due to the non-IID distribution, participants tend to drift towards the minima of the local objective functions and, consequently, model updates become significantly different across participants, hindering model convergence. In addition, previous works (Stich, 2018; Wang and Joshi, 2018) have shown that there exists an error term that monotonically increases with each local step and is exacerbated with non-IID data. This error can be negligible when the learning rate decays. As learning rate decay skips spurious local minima by starting from an initial large learning rate, it benefits convergence by avoiding oscillation with the decayed learning rates. Thus, a learning rate decay of 0.1 was added to our scheduler.

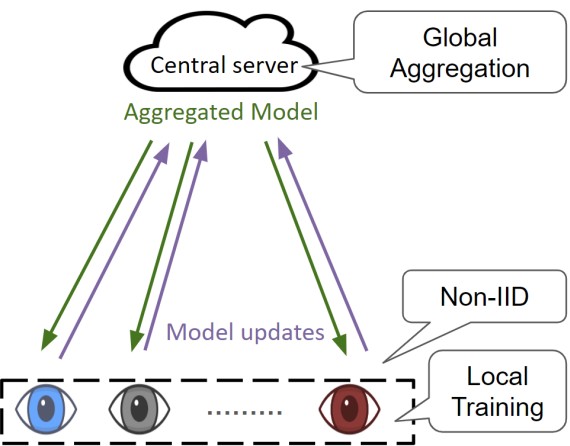

Figure 1: Cross-device centralised horizontal FL

### 3.4. Adaptive Federated Learning for Gaze Estimation

Due to the heterogeneity of in-the-wild gaze data, training a single global model across users is known to result in lower performance due to having different local updates across participants. Therefore, simply averaging the updates (i.e. FedAvg (McMahan et al., 2017)) leads to increasing the error rate even with learning rates or learning rates decay. Alternatively, the expectation of the aggregated local updates should follow the global objective.

In this work, for each communication round $t$, the model is optimised on two levels. While the participant optimiser aims to minimise Equation 2 according to the local data of each participant, the server optimiser aims to optimise it globally on the aggregated model. In other words, the participant's updates are treated as pseudo-gradients to aggregate the model on the server side while adapting to the heterogeneous local updates, which result from the heterogeneous distribution by incorporating knowledge from previous iterations for better-informed optimisation, as shown in algorithm ?? (lines 18-23). Specifically, the server gathers the local differences $\nabla_i^t$, averages them into the pseudo-gradient $\nabla_t$, and updates the aggregated model using the Adam optimiser.

Although adaptive optimisation methods (e.g. Adam (Kingma and Ba, 2014; Zaheer et al., 2018)) have proven their effectiveness in training deep neural networks, improving the performance of SGD, and guaranteeing convergence, especially for gaze estimation tasks (Hansen and Ji, 2009; Zhang et al., 2017a,b), SGD maintains the same computation and communication costs on the participant side and does not depend on the participant sampling ratio. Here, the participant and server optimisers are SGD and Adam, respectively. This combination of optimisers has been proven to be effective both theoretically and empirically (Wang et al., 2021).

Additionally, for robustness, we investigate the performance of our approach against more heterogeneous setups (e.g. poisoning attacks, outliers, etc.). As a simulation, Gaussian noise was added with a standard deviation of 0.5 to the participants' training data, according to (Mills et al., 2021).

---

**Algorithm 1** Adaptive FL Algorithm

---

1: **The central server:**
2: Initialises model parameters $w_0$ and broadcasts them to all participants.

3: **for** each global model update round $t = 1, 2, ..., T$ **do**
4:     The server determines $C_t$, which is the set of randomly selected participants.
5:     **for** each participant $i \in C_t$ **in parallel do**
6:         Obtain the latest model parameters from the server, i.e. set $w_i^{t,0} = w_t$.
7:         **for** each local epoch $e = 1, 2, ..., E$ **do**
8:             batches $\leftarrow$ randomly divide dataset $D_i$ into batches of size $M$.
9:             Obtain the local model parameters from the last epoch, i.e. set $w_i^{t,e} = w_i^{B,e-1}$.
10:             **for** batch index $b = 1, 2, ..., B = \frac{n_i}{M}$ **do**
11:                 Compute the batch gradient $g_i^b$ using the participant optimizer in 1.
12:                 Update model parameters locally: $w_i^{b+1,e} \leftarrow w_i^{b,e} - \mu g_i^b$.
13:             **end for**
14:         **end for**
15:         Send the updated model $\nabla_i^t = w_{i,E}^t - w_t$ to the central server.
16:     **end for**
17:     **Server optimiser:**
18:     The central server aggregates the received model weights:
19:     $\nabla_t = \frac{1}{|C|} \sum_{i \in C} \nabla_i^t$
20:     The central server optimizes the model with its learning rate $\beta$:
21:     $a_t = \beta a_{t-1} + (1 - \beta) \nabla_t$
22:     $v_t = \beta v_{t-1} + (1 - \beta) \nabla_t^2$ (Adam)
23:     $w_{t+1} = w_t + \eta \frac{a_t}{\sqrt{v_t} + \tau}$ where $\tau$ is the degree of adaptivity
24:     The central server checks if the maximum number of rounds is reached.
25:     The central server broadcasts the aggregated model parameters to all participants.
26: **end for**

---

## 4. Results

As the participants' population can rapidly evolve in gaze estimation applications, non-IID distribution shifts arise and in-the-wild gaze challenges intensify. Thus, for a better understanding of the model robustness and the heterogeneous updates across participants, we first evaluate our approach and the baselines under person-specific and person-independent evaluation schemes. The former holds out a fraction of the data of each participant for validation to ensure that the validation set is representative and to measure the local performance, while the latter leaves one participant out of the training set for evaluation of the global model generalisation capability. We perform our evaluations on the MPIIGaze dataset (Zhang et al., 2017b) in terms of the mean angle error, i.e. the difference between the predicted gaze angles and the dataset ground truth.

**Dataset** The MPIIGaze dataset contains 15 participants with 213,659 annotated data samples collected over 9 days to 3 months with varying head poses, gaze targets, and illumination conditions. The number of images collected by each participant varies between 1,498 and 34,745. Consequently, the dataset was chosen due to its daily real-life representation of different users, environments, and cameras, emphasising the heterogeneous in-the-wild data distribution among different participants.

**Baselines** We compare our method against three baselines: An individual learning scheme, a data centre learning scheme, and a conventional federated learning approach.

- **Individual Learning:** Each participant $p_i$ trains the model locally on the local dataset $D_i$. The data is neither shared with other participants nor with a central server. However, the model is only exposed to a relatively small amount of samples which, consequently, yields a poor generalisation performance on out-of-the-training-set samples, lacks robustness and, therefore cannot be deployed to other participants due to the non-IID data distribution.

- **Data Centre Learning:** Since the availability of a large number of training samples is a stepping stone for model training, data centre learning collects data samples from all participants and trains the model on one central dataset. Therefore, the model generalisation performance increases but participants' data is shared with the central server and privacy leakage risks arise.

- **Federated Average (FedAvg):** We compare our approach with the conventional federated learning setting, FedAvg, where the server aggregates the model updates by simply averaging them.

**Person-Independent Performance** For individual learning, the results in Figure 2 show that the gaze estimator yields a poor generalisation performance on out-of-the-training-set samples (person-independent), lacks robustness and, therefore, cannot be deployed to other participants due to the non-IID data distribution. For data centre learning, despite the increase in model robustness shown in Figure 3, participants' data is shared with the central server and privacy leakage risks arise. Finally, for federated learning, we employ the same number of communication rounds and computations to our adaptive framework and vanilla FL (FedAvg) for a fair comparison. Instead of performing a number of steps in FedAvg, we perform a fixed number of epochs for each participant to accommodate the data imbalance

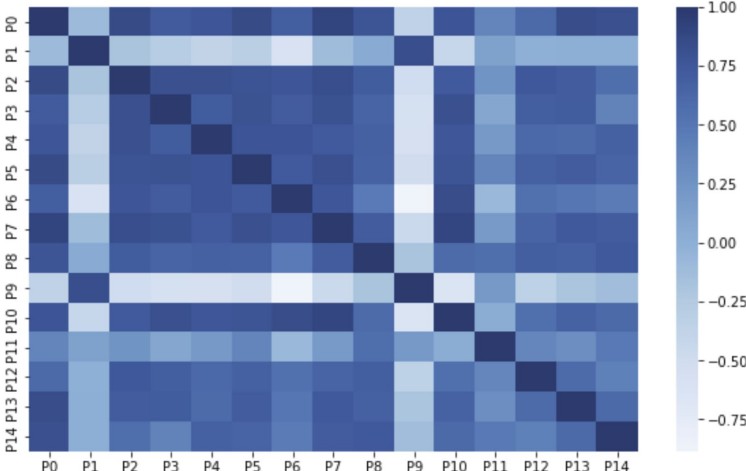

Figure 2: Person-specific (diagonal) and person-independent mean angle error for individual training

of the local datasets and to avoid training on the same data sample repeatedly. Figure 3 shows that, in FedAvg, convergence issues occur. As shown in Figure 3 and Figure 4, in user-independent setups, our adaptive FL approach outperforms FedAvg by 15.8% (from a mean error of 10.63 degrees to 8.95 degrees) and achieves convergence while maintaining the same communication cost as FedAvg. It also outperforms individual training (from a mean error of 9.10 degrees to 8.95 degrees) and can potentially improve with more participants in practical setups. Finally, the performance gap between our approach and the data centre training comes at the cost of privacy and scale (see section 5) and can potentially be bridged in practice with more participants and training rounds over time.

**Person-Specific Performance**   The local model performance is essential for the adaptivity of the gaze estimator to the divergent model updates, as well as to ensure fairness by correcting the unintended behaviours exhibited by the gaze estimator in heterogeneous collaborative setups. In the FL literature, fairness is measured by the minimum performance metric across all clients. As shown in Figure 3, FedAvg resulted in a mean angle error of 9.3 degrees and 12.2 degrees for the minimum and maximum participant performance, respectively, while our approach resulted in 7.5 degrees and 10.2 degrees.

**Robustness**   Due to the open nature of FL and the involvement of a large number of participants, the training process can be vulnerable to failures (e.g. distribution shifts and data poisoning). As distinguishing heterogeneous distributions from poisoning attacks can be challenging, we test our model on noisy participants to check the model's robustness, i.e. the ability of the gaze estimator to function correctly in the presence of such problems. Results, in Figure 4, show that the model is robust to $\geq 70\%$ of noisy participants and it is important to mention that the effective rate of malicious activity or outliers can be extremely small in cross-device settings, where thousands of participants are involved (Wang et al., 2021).

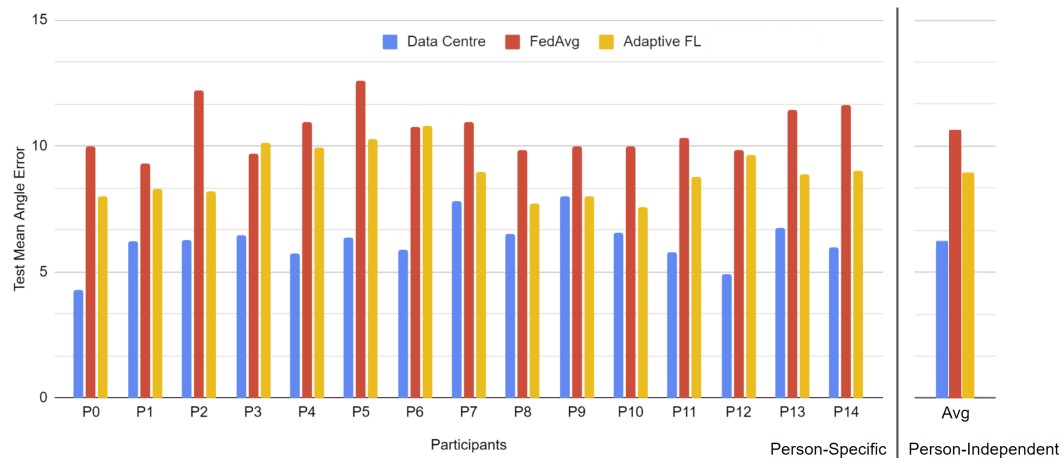

Figure 3: Mean angle error for person-specific evaluation for each participant (left) and the average person-independent evaluation (right)

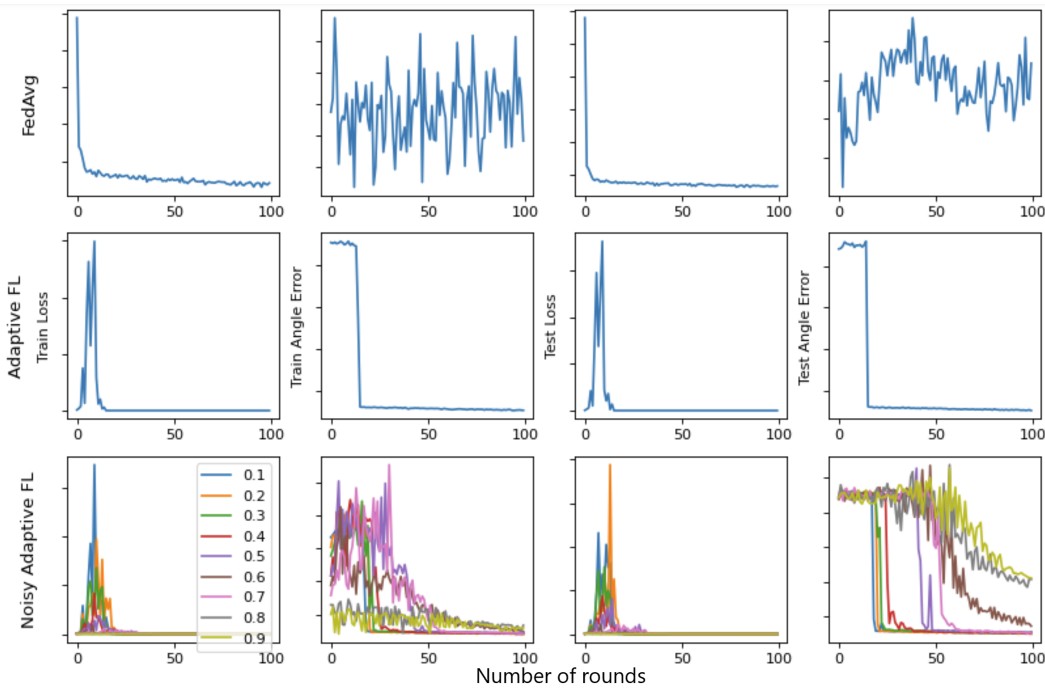

Figure 4: The training and test losses and mean angular errors for FedAvg (first row), our approach (second row), and robustness to noisy participants (third row)

## 5. Discussion

Our work paves the way to developing privacy-preserving collaborative learning setups for gaze estimation while maintaining the utility of the model. FL extends prior non-federated gaze estimation directions along with other unique approaches to address privacy, fairness, and robustness.

**Privacy of Gaze Data**  Our FL approach is able to enhance data privacy via data minimisation. It focuses on reducing the attack surface on gaze data by sending the model minimal-focused updates needed for the gaze estimation task instead of the raw data while maintaining a better model performance than FedAvg. It also yields a minor loss in performance compared to the non-private data centre training, as shown in Figure 3, since the model is no longer directly trained on the entire dataset but instead, the individual model updates are combined by the aggregation function. In addition, our work presents a privacy-preserving solution that does not interfere with the user experience as it requires the same computation and communication costs as FedAvg on the user side.

**Gaze Data Heterogeneity**  Deep learning models for gaze estimation often exhibit unintended behaviours that lead to undesirable performance due to human, environmental, and device differences resulting in data heterogeneity. Fairness concerns about gaze estimation models can be exacerbated in FL settings due to data heterogeneity. We evaluated the adaptivity of our work under person-specific, person-independent, and fairness metrics. Our results show that our approach indeed adapts the model updates to the participants' data distribution resulting in better performance, as shown in Figure 3. We also investigated model robustness against random noise, as shown in Figure 4. Nonetheless, attacks on gaze estimation models can be further studied.

**Limitations and Future Work**  Apart from the privacy guarantees presented in this paper, a limitation of our work resides in the fact that FL does not provide provable (cryptographic) privacy guarantees. Thus, a direction for further research could be to combine FL with formal privacy technologies (e.g. differential privacy and secure aggregation (Bonawitz et al., 2017)). In addition, as we introduce additional communication through FL in comparison to conventional gaze estimation approaches, allowing participants to join the federation when they are idle (i.e., charged and connected to wi-fi, or during local nighttime) could potentially enhance the user experience in practical settings. Furthermore, applying these techniques for high-dimensional and large-scale gaze estimation applications under in-the-wild non-IID distributions, and developing better evaluation metrics that cover both the person-specific and person-independent performance for federated learning local and global models, remain open research directions. Lastly, finding the right balance between fairness, robustness, and privacy is a challenging task. While fairness ensures that the model performs well for participants with different data distributions, robustness ensures that the outliers do not affect the model performance, and privacy ensures that the model does not retain information about outlier participants' data. However, adaptive learning and personalisation techniques are believed to balance this interplay (Jiang et al., 2019; Wang et al., 2019), and could be further studied with our approach.

## 6. Conclusion

In this paper, we presented a federated learning approach for gaze estimation that allows model training to adapt to the divergence of participants' heterogeneous data while alleviating privacy concerns of data sharing. To the best of our knowledge, this is the first work that integrates federated learning with gaze estimation tasks. We showed that our approach enhances the privacy guarantees of conventional methods, handles the convergence of gaze estimators, and significantly outperforms the performance of vanilla federated learning. We believe that our work paves the way to developing privacy-preserving collaborative learning setups for gaze estimation tasks while maintaining a good model performance.

## Acknowledgments

M. Elfares was funded by the Ministry of Science, Research and the Arts Baden-Württemberg in the Artificial Intelligence Software Academy (AISA).
Z. Hu was funded by the Deutsche Forschungsgemeinschaft (DFG, German Research Foundation) under Germany's Excellence Strategy - EXC 2075 – 390740016.
R. Küsters and P. Reisert were supported by the German Federal Ministry of Education and Research under Grant Agreement No. 16KIS1441 and from the French National Research Agency under Grant Agreement No. ANR-20-CYAL-0006 (CRYPTECS project).
A. Bulling was funded by the European Research Council (ERC; grant agreement 801708).

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
