# OpenReview forum: "Federated Learning for Appearance-based Gaze Estimation in the Wild"
_NeurIPS.cc/2022/Workshop/GMML — Gaze Meets ML 2022 Poster_

### Official Review · Reviewer_Qnvk · 2022-10-16
**A very interesting paper, with some interesting ideas involving federated learning for gaze estimation in a distributed system.**

**Rating:** 7
**Confidence:** 3

**Review:**

The current paper "Federated Learning for Appearance-based Gaze Estimation in the Wild", is proposing an interesting approach, namely the introduction of federated learning in gaze estimation to protect the subjects' privacy.

While the authors touch quite a few subject introducing the topic (see Introduction) and the existing solutions (see Related Work) the reviewer got the impression that the authors omit to explain what are the real problems (see privacy) and how this federated learning work and more importantly, there is no real conclusion what the current solutions can provide, what are really their limitations and how those limitations can be solved by the current strategy (see proposed in the manuscript). I would kindly recommend to focus more on this side of the paper. This part looks very superficial.

Apparently, the authors are using the CNN architecture considered by Zhang et al, however there is no sign of any description whatsoever how that network looks like, why this network architecture is appropriate and how they can profit out of this model. Many of the parameters appear as "magic numbers" and not explained in any way (see learning rate, momentum, C, etc.)

As it is mentioned, the training is happening locally, each participant training with their respective data, and the weights (updates) are submitted to the server which averages the model parameters and sends back the aggregated models updates to the participants. This seems to a logical strategy, but are these participants (not the participant but their data) "equivalent"? I means size wise (see unbalance issue, quality wise, etc.) Such artifact can be absorbed by just simply averaging the values.

As it seems this issue (see above) is tackled by a server side optimization, which is appropriate, and the results (see Fig.3) show the necessity for such server side optimization of the features.

The Alg. 1 description is proper and useful, however, it would be advantageous to have an overall system overview for the reader to understand how the data is going, who is doing what, etc.

The comparison with individual learning, data centre learning and federated averaging are very useful and the result obtained by the Adaptive FL they are way better than the Avg FL suggested in earlier works.

In Fig 4 the training part is not telling much, but definitely it is worth mentioning it.

Altogether, the paper is well written, despite the lack of the model description, the results are good and this should be appreciated. As it is claimed it looks like to be the first work in gaze estimation when federated learning was involved.

---

### Official Review · Reviewer_UH46 · 2022-10-17
**Training a gaze model with a federated learning approach.**

**Rating:** 7
**Confidence:** 3

**Review:**

I like this paper as a way to address concerns people have with personal data gathering. It would be nice to call out explicitly the results in Figure 3 to help people understand the cost of using a federated learning approach (e.g., 9 degree error vs 5 degree or what not) with some insight into if larger datasets for training could reduce this (basically making the argument that federation will allow gathering of larger data sets which would ultimately result in better models - if this is true).

A few specific comments:
* i.e. should be followed with a comma
* Is the model topology shared among the federation always the same, or does it evolve based on information gathered? If the latter would you expect multiple back and forths to retrain the weights on different topologies?
* Is gaussian noise a good model for attacks? If I wanted to really confuse the model would I not just train for the reverse of the correct answer? Could you detect this sort of thing?

---

### Official Review · Reviewer_aqea · 2022-10-17
**Federated learning shows merit for gaze tracking**

**Rating:** 6
**Confidence:** 3

**Review:**

**Summary**

This study applies two federated learning approaches on the appearance based gaze estimation problem. This approach to this particular problem seems well merited as privacy concerns are obvious for user video. Performance of the ADAM model shows significant improvement over simply averaging local gradients (FedAvg), but leaves a large gap relative to using the entire dataset. This work presents a useful test case for federated learning.

**Originality**
- Addressing privacy concerns for personal data is important from an ethical perspective as well as having the potential to allow much more powerful models to be trained if access to larger datasets becomes possible. It does not appear that federated learning has been applied to this particular problem, and seems to be a more practical solution in comparison to adding noise.

*Weaknesses*
- The utilized algorithm does not differ from the ones proposed by reference [35]. This is fine but it would be nice to see the exploration of meta-parameters expanded beyond the single models utilized here.

**Clarity**
- The models presented here and the motivation are well presented.

*Weaknesses*
- The authors miss an opportunity to explain the model itself and differences between models (particularly the variants of federated learning) graphically. As is Figure 1 provides very little information and could be reformulated to express the models in more detail.
- Figure 2 shows the person specific model performance. It is not clear why this figure and figure 3 both have the y/color axis referring to “mean angle error” but figure 2 seems to scale between 1 and -1. It is additionally unclear why the diagonal appears to be all 1s. It seems some form of normalized error is instead being presented but this is not clearly explained. It is also not completely clear how train-test splits are performed for this analysis. The individual learning condition seems particularly interesting and it should be expanded upon for comparison with federated learning.
- Figure 4 lacks y-values on all panels. Additionally the scales are not shared across rows so models cannot be compared in any meaningful way.

**Significance**
- This work seems to hold particular interest for anyone interested in real world applications of gaze tracking.

*Weaknesses*
- The authors leave any analysis of the true privacy implications (does federated learning actually preserve privacy) to future work.

- They authors do not explore any additional constraints proposed by real world data generation: eg. heterogeneity in dataset size, reliability, etc. Expanding on one of these topics would greatly increase the value of this work. While they explore adding gaussian noise to a subset of the data, they do not explore how the amount of data available at each local device should impact model performance. This form of data heterogeneity would seem to be of near equal relevance as the variability in data distribution across individuals. The broader significance of this work would be strengthened by including this or a similar analysis.

**Questions**
- How does the federated model performance on held out data compare to a model only trained on an individual participant? Depending on one’s objective, the value of the federated training framework would depend greatly on the federated model outperforming a model only trained locally on that individual’s data. This does not seem to be explored in this study but including this would be very valuable.

- In a hypothetical ‘real world’ dataset labels would not exist. While addressing unsupervised or semi-supervised approaches feels beyond the scope of this work, these limitations should be addressed and potential solutions considered. Is manual annotation completely incompatible with the federated framework (I assume so)?

---

### Meta-Review · Area_Chair_QEdS · 2022-10-20

**Recommendation:** Accept (Poster)
**Confidence:** 4

**Metareview:**

This work introduces federated learning approaches in gaze estimation. All reviewers noticed the originality of the work and the solid experiments. The paper is also clear in terms of its presentation, albeit there is room for improvement, e.g., further explaining the importance and limitations of federated learning for privacy protection in this particular domain. As mentioned by the reviewers, the work could be of interest to the general workshop audience, and the positive comments suggest acceptance. There are several questions raised, as well as suggestions for improvement, that the authors are recommended to consider when revising the camera-ready version.

---

### Decision · Program_Chairs · 2022-10-20

Accept (Poster)